# Sleep Stage Classification from Wristband Sensor Data in Patients with Sleep Disorders

Oriella Gnarra[1,2], Camilla Massaro[1,2], Jan D. Warncke[3], Tobias Nef[3,4],
Markus Schmidt[3,5] and Diego Paez-Granados[1,2]

*Abstract*—Polysomnography (PSG) is the clinical gold standard for sleep assessment, but it is constrained to single-night recordings in hospital environments and requires trained personnel. The emergence of wearable technologies enables unobtrusive and potentially long-term sleep monitoring. This study presents an AI-based approach tailored to individuals with sleep disorders, evaluating the U-Sleep neural network trained on multimodal signals: Acceleration (ACC), Blood Volume Pulse (BVP), Electrodermal Activity (EDA), and skin Temperature (TEMP), collected via the Empatica E4 wristband. The goal is to advance precision health by developing sleep monitoring tools adaptable to individual physiological profiles, particularly in patients with suspected or diagnosed sleep disorders. This is the first study of multimodal sleep stage classification combining EDA and TEMP with ACC and BVP, which was developed using data from 127 participants undergoing simultaneous PSG and wearable recordings. Evaluating and verifying an improved performance through the integration of diverse physiological signals in a personalized model for individuals with sleep disorders. Bland-Altman analysis revealed an overestimation of Light sleep and an underestimation of Sleep Onset Latency (SOL). However, sleep efficiency, REM latency, wake after sleep onset, REM, and Deep sleep durations were estimated without significant differences from PSG. Epoch-by-epoch accuracy reached 0.87±0.07 for Wake, 0.90±0.04 for REM, 0.71±0.07 for Light, and 0.89±0.04 for Deep sleep. Overall accuracy and F1-score were 0.69±0.08 and 0.62±0.11 for patients and 0.77±0.05 and 0.74±0.06 for healthy participants, respectively. These results highlight the feasibility of applying multimodal wearable data for accurate sleep staging across diverse populations. By enabling models to capture individual-specific patterns, this work contributes to the field of precision sleep health and lays the groundwork for remote, patient-tailored management of sleep disorders. Although further refinement is needed before clinical application, the approach shows promise for advancing AI-driven, personalized diagnostics in real-world settings.

*Index Terms*—Data Analysis, Deep Learning, Sleep Stages, Sleep Disorders, Wearables.

## I. INTRODUCTION

Sleep is a biological process in which mental and physical activity is decreased. On average, it constitutes approximately one-third of a human lifetime, highlighting its fundamental role in overall well-being. The way a person sleeps influences many aspects of their life, such as attention, memory, and mental and physical health. Lack or poor quality of sleep can increase the risk of cardiovascular and metabolic diseases, as shown by Zheng et al. [1], lead to cognitive deficits or mental health problems, and cause accidents. Moreover, recent studies have shown that reduced levels of slow wave sleep and Rapid Eye Movement (REM) sleep are correlated to brain atrophy [2]. For this reason, monitoring sleep and being able to classify its stages is fundamental to identifying and managing these problems by providing important biomarkers for sleep disorders. PSG is the clinical gold standard for sleep measurement, but is limited to single-night hospital settings with trained personnel, requiring doctors to manually score sleep stages, looking at the collected parameters. A known limitation of manual sleep stage scoring is variability in stage classification when identical data sets are scored by different raters. The inter-rater agreement for the four-stage sleep classification is 83.7% according to Nikkonen et al. [3], which considers 10 sleep experts scoring from 7 different sleep centers, including 50 PSG recordings from both healthy participants and sleep-disordered patients. Moreover, PSG consists of multiple types of electrodes attached to various parts of the body, which may cause unnatural sleep for the patient [4]. This results in obtrusive, time-consuming, and expensive sleep monitoring. Recent research increasingly focuses on developing more accessible and practical solutions for sleep monitoring, to accelerate the detection of sleep disorders and facilitate the study of their associations with other health conditions [5]. The growing demand for home-based, unobtrusive sleep assessment has led to the widespread adoption of wearable and nearable technologies—including smartwatches, wristbands, rings, and sensor-embedded mattresses [5]. These devices, which span both consumer-grade and research-grade products, are capable of capturing various physiological signals such as movement, Heart Rate (HR), EDA, and peripheral oxygen saturation, often at different levels of resolution. In addition, others are equipped with proprietary algorithms that estimate sleep stages and compute sleep-related metrics such as Total Sleep Time (TST), Sleep Efficiency (SE), SOL, and Wake After Sleep Onset (WASO) [6]. Despite their convenience and scalability, a major challenge remains: improving the accuracy and clinical reliability of sleep detection and staging performed by these devices. This concern is supported by findings from a prospective multicenter validation study, which evaluated 11 commercially available sleep trackers and reported substan-

*The corresponding author is oriella.gnarra@hest.ethz.ch.
Markus Schmidt and Diego Paez-Granados share the last authorship.
[1]SCAI Lab, Department of Health Sciences and Technology, ETH Zurich, Zurich, Switzerland.
[2]Swiss Paraplegic Research (SPF), Nottwil, Switzerland.
[3]Sleep-Wake Epilepsy Center, Department of Neurology, University Hospital of Bern, Bern, Switzerland.
[4]Gerontechnology Rehabilitation Group, ARTORG Centre for Biomedical Engineering Research, University of Bern, Bern, Switzerland.
[5]Ohio Sleep Medicine Institute, Dublin, Ohio, USA.

This research project was partially supported by the Interfaculty Research Cooperation "Decoding Sleep" of the University of Bern and the JST Moonshot R&D Program, Grant Number JPMJMS2034-18.

tial variability in performance, with most devices showing limited agreement with PSG for sleep stage classification [7]. Moreover, many current systems lack rigorous validation against PSG, particularly in populations with disrupted or atypical sleep. There is also a growing need to develop models that can take full advantage of the multimodal data these devices provide, potentially unlocking novel insights into sleep physiology while maintaining practicality for everyday use. Some studies have attempted sleep staging using only a single modality, such as EDA [8], or combinations of ACC and Photoplethysmography (PPG) [9], highlighting both the promise and the limitations of unimodal or bimodal approaches.

The main goal of this study is to classify sleep stages using wearable vital monitors, exploiting the non-linear modeling capabilities of deep neural networks trained on data from individuals with diagnosed or suspected (normal variants) sleep disorders. Unlike previous work, primarily focused on healthy populations, our approach centers on evaluating wearable-based sleep staging in clinically relevant individuals characterized by heterogeneous, often disrupted sleep patterns. This distinction aligns with the core principles of precision health, where models and tools must adapt to the specific needs of diverse patient groups. By leveraging multimodal sensor data, including ACC, BVP, EDA, and skin TEMP, we investigate the potential of Artificial Intelligence (AI) to learn individual-specific physiological patterns that reflect distinct sleep architectures.

## II. METHODS

### A. Dataset

The dataset analyzed for this paper is part of the Sleep Monitoring Study (SMS) [10]. Before the start of the study, approval was granted by the Berne Cantonal Ethics Committee (Basec 2021-00461), and written consent was obtained from all participants. One hundred and twenty-seven participants were recorded for one night in an examination room at the Sleep-Wake Epilepsy Centre (SWEZ) of the University Hospital Bern, Switzerland. Demographic information is presented in Table I. The most common diagnoses in the cohort were sleep-related breathing disorders, such as obstructive sleep apnea, and central disorders of hypersomnolence, including narcolepsy with or without cataplexy. These were followed by insomnia, parasomnias (such as REM sleep behavior disorder), and sleep-related movement disorders, such as periodic limb movement disorder. The clinical dataset contains missing diagnostic information and includes participants categorized under other or unspecified sleep disorders. The term normal variant is clinically applied to individuals exhibiting non-pathological sleep patterns. The entire dataset was divided into training, evaluation, and test sets. The balance between the different diagnoses was maintained for each subset. The participants' sleep was monitored simultaneously with a wristband, Empatica E4, placed on the non-dominant arm and PSG SOMNOscreen® HD Somnomedics. The data provided by the E4 are a measurement of the BVP at a sampling frequency of 64 Hz calculated by the PPG; the ACC in the range [-2 g, 2 g] at a sampling frequency of

32 Hz calculated by a 3-axis accelerometer; the skin TEMP with a sampling frequency of 4 Hz calculated by an infrared thermopile; and the EDA at a sampling frequency of 4 Hz calculated by a perspiration sensor. These signals, represented in Figure 1, are input for the deep neural network for sleep stage classification. The ground truth labels are provided by the PSG scoring within the clinical routine.

TABLE I: Demographic characteristics of participants in the subsets used for model training, evaluation, and testing.

| Dataset | Full | Train | Eval. | Test |
|---|---|---|---|---|
| Participants (n) | 127 | 86 | 15 | 26 |
| Females (%) | 54.89 | 56.04 | 53.33 | 51.85 |
| Age (years) | 45.3±16.2 | 46.2±16.4 | 44.6±15.7 | 42.2±16.1 |
| **Diagnosis (%)** | | | | |
| Breathing disorders | 59.1 | 60.5 | 60.0 | 53.9 |
| Hypersomnolence | 15.0 | 14.0 | 13.3 | 19.2 |
| Insomnia | 5.5 | 5.8 | 6.7 | 3.9 |
| Missing diagnosis | 4.7 | 4.7 | 6.7 | 3.9 |
| Parasomnias | 4.7 | 4.7 | 6.7 | 3.9 |
| Normal variants | 4.7 | 3.5 | 0.0 | 11.5 |
| Other disorders | 4.7 | 5.8 | 6.7 | 3.9 |
| Movement disorders | 1.6 | 1.2 | 0.0 | 3.9 |

Age is expressed in mean ± Standard Deviation (SD).

### B. Data Processing

The wearable data were trimmed between the PSG lights-off and lights-on times [11]. ACC and BVP data were preprocessed respectively with median centering and z-score normalization (i.e., zero mean and unit variance). Both signals then underwent adaptive Interquartile Range (IQR) normalization using a 300-second sliding window, followed by outlier clipping at thresholds of 20 times the IQR [9].

EDA was processed following the approach in [12], where the raw signal was decomposed into tonic and phasic components, retaining only the phasic component for further analysis. The phasic component of EDA is directly proportional to theta waves, dominant in light sleep, and inversely to delta waves, dominant in deep sleep [13]. TEMP was transformed by computing the first-order difference (delta) between consecutive samples, which was used in place of the raw temperature signal [14]. These transformed EDA and TEMP signals were then normalized, using median normalization, adaptive IQR normalization (300-second window), and outlier clipping at 20× and 15× IQR, respectively.

To ensure temporal alignment across modalities, all signals were resampled to a uniform frequency of 32 Hz. This involved downsampling PPG and upsampling EDA, TEMP. Finally, each signal was converted into a spectrogram representation using the Short-Time Fourier Transform (STFT) to provide the model with time-frequency domain input.

### C. Deep Neural Network Architecture

The chosen architecture is based on U-Sleep [15], a fully convolutional network based on U-Net [16], [17], which is a state-of-the-art algorithm for classifying sleep stages from Electroencephalogram (EEG) and Electrooculogram (EOG). The network can be divided into four parts: the conformation module, which prepares the input for the network by reshaping

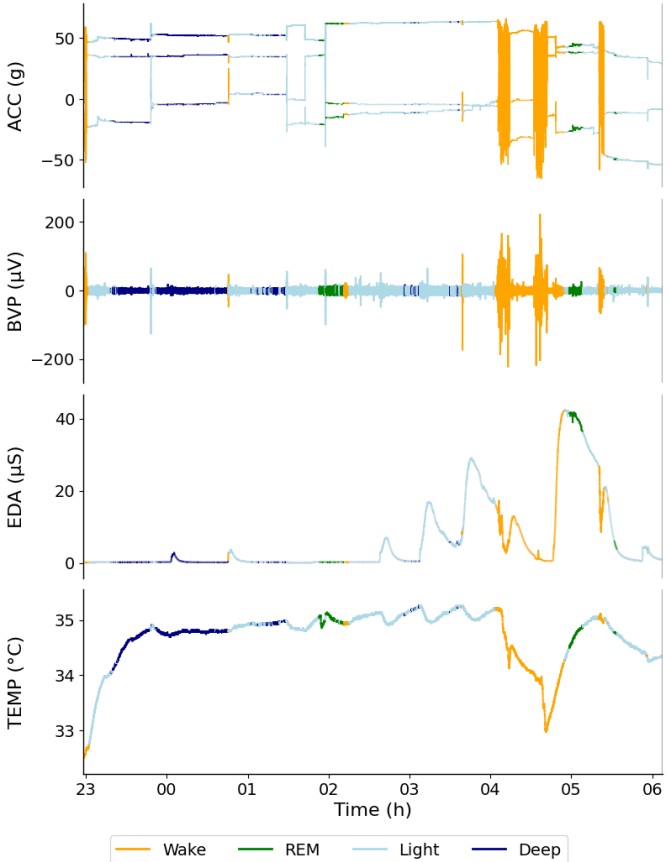

Fig. 1: Multichannel physiological signals from one participant with obstructive sleep apnea recorded by the wristband, color-coded by sleep stages.

and zero padding it; the encoder module, which extracts features from the data, compressing it to a lower temporal dimension; the decoder module, which brings the compressed feature representation of the input back to the initial temporal dimension; and the segment classifier module, which segments and classifies the decoded vector into single epochs.

The conformation module has been introduced in [9], while the other modules were already present in U-Sleep [15]. Olsen et al. [9] clearly show the network architecture used in our work. This model was selected because it had been previously tested in another wearable device [9], which made slight modifications from [15] and leveraged ACC and PPG signals, both of which are also present in Empatica E4. Furthermore, this model was preferred over the Convolutional Neural Network (CNN) previously tested on the E4 device in [18] due to its utilization of state-of-the-art techniques and flexibility to accommodate any type and number of input time series data. We specifically chose a U-Net architecture for sleep stage classification because it is well-suited for time series segmentation tasks, where temporal resolution and local context are critical. U-Net's encoder-decoder structure enables the model to capture both short-term patterns and long-range dependencies by combining multi-scale feature representations. This is particularly crucial in this context, where transitions between stages often depend on complex temporal

patterns across different physiological signals. While recent approaches based on attention-based architectures and Large Language Models (LLMs) have shown promise in sequential data tasks [19], they typically require large-scale pretraining on massive datasets and significant computational resources, which are often impractical in clinical or wearable settings. In contrast, our chosen U-Net-based model can be trained end-to-end on relatively small datasets, making it more practical and better aligned with the real-world constraints of sleep monitoring in patients with diverse disorders using wearable devices. The main contributions of our work include adapting the network presented in [9], experimenting with various signal combinations, identifying the optimal parameters, and conducting a comprehensive model evaluation similar to that performed on the other devices. The initial study [9] utilized ACC and PPG, data gathered from a consumer wearable device. Empatica E4 additionally offers various other signals, including EDA and skin temperature, as well as derived measurements, such as HR and Inter-Beat Interval (IBI), which are derived from PPG. In our work, we evaluated different combinations of input signals. The final model was trained using a multimodal input composed of ACC, PPG, EDA, and TEMP.

*D. Evaluation*

The dataset was split stratifying it per diagnosis, as shown in Table I, dividing it into training (65%), evaluation (15%), and test set (20%). To avoid data leakage, we ensured that all segments belonging to the same patient were assigned exclusively to one of the training, validation, or test sets. This way, the model was evaluated only on data from patients it had never seen during training. The data were divided into consecutive, non-overlapping windows, each containing a fixed number of 30-second epochs, to ensure uniform input sizes. After computational experiments, the input segment size was chosen to be 1024 epochs, which corresponds approximately to 8.5 hours of sleep, covering, in most cases, the whole night of sleep. Recordings shorter than the input size were zero-padded, while the few longer ones were cut to 1024 epochs. A binary focal loss was used, dynamically adjusting class weights based on batch distribution. For each batch:

$$L = -\frac{1}{N} \sum_{i=1}^{N} \left[ (1-p_w)^\alpha \mathbf{1}_{y_i=1} \log(p_i) + p_w^\alpha \mathbf{1}_{y_i=0} \log(1-p_i) \right]$$
(1)

where: $p_i$ is the predicted probability for sample $i$; $y_i$ is the true label for sample $i$ (0 or 1); $p_w$ is the positive sample ratio in the batch; $\alpha$ is the focusing parameter on hard examples; and $\mathbf{1}(\cdot)$ is the indicator function.

Hyperparameter tuning was performed using HyperBand [20] to find the model's best parameters on the evaluation set. The parameters to optimize were: $M$, which is the number of encoders and decoders, $K$ which is the kernel height of the 2D convolutions, and initial filters used in the 2D convolution of the first encoder, the focusing parameter $\alpha$, and the learning rate for the loss, minimized with ADAM optimizer [21]. The evaluation metrics were chosen to compare results with State Of The Art (SOTA) and account for the imbalanced dataset. For the first scope, accuracy was used, both as a per-class

metric and as a global metric. The other metrics used are on a single-stage level: sensitivity, specificity, F1-score, recall, and precision, and on the global level, balanced accuracy, macro-F1, and Cohen's $\kappa$. Sleep measures were also evaluated, including TST, SOL, REM Latency (RL), SE, $\text{WASO}_d$, REM sleep duration ($\text{REM}_d$), light sleep duration ($\text{Light}_d$) and deep sleep duration ($\text{Deep}_d$).

## III. RESULTS

Hyperparameter tuning identified the optimal configuration as $M = 10$, $K = 16$, initial filters = 32, $\alpha = 0.15$ and learning rate = $10^{-3}$. Some performance analysis for this configuration is shown in this section, following the standardized framework presented in [11]. All experiments were conducted using Python 3.9.13 with TensorFlow 2.11.0 and a range of scientific computing and signal processing libraries. Analyses were performed on a remote Jupyter server equipped with an NVIDIA GPU supporting CUDA for accelerated computation of deep learning tasks. Each signal modality, after being preprocessed, was provided as input to the CNN architecture, which integrates multi-source information for classification.

### A. Sleep measures analysis

In Table II, a comparison of sleep measures is made between PSG and our model. Mean values and SD, and p-values from a paired t-test if the differences follow a normal distribution, otherwise from the Wilcoxon signed-rank test, are shown for each sleep measure. The analysis revealed an underestimation of SOL and an overestimation of Light sleep. However, TST, RL, SE, WASO, REM, and Deep sleep durations were estimated without significant differences from PSG. In Figure 2, Bland-Altman plots are shown for each sleep measure. The plots show the mean on the x-axis and the differences on the y-axis expressed in minutes between the sleep measures of our model and PSG, except for SE, which is expressed as a percentage on both axes.

TABLE II: Mean difference between PSG and our model for sleep measures classification.

| Sleep Measure | PSG | Model | p-value |
|---|---|---|---|
| TST | 359.23 ± 78.41 | 372.15 ± 86.50 | 0.24 |
| SOL | 18.92 ± 15.15 | 12.81 ± 12.29 | 0.02* |
| RL | 132.78 ± 76.46 | 107.75 ± 72.49 | 0.12 |
| SE | 74.22 ± 13.82 | 76.25 ± 12.78 | 0.20 |
| $\text{WASO}_d$ | **60.83 ± 41.40** | **54.81 ± 42.53** | **0.55** |
| $\text{REM}_d$ | **60.90 ± 33.09** | **57.15 ± 30.52** | **0.55** |
| $\text{Light}_d$ | 223.25 ± 60.86 | 245.60 ± 61.84 | 0.04* |
| $\text{Deep}_d$ | 75.08 ± 31.79 | 69.40 ± 32.37 | 0.39 |

Comparison of sleep measures (mean and SD) between PSG and our model on the test set. $p < 0.05*$ represents a significant difference between the ground truth and our model. The measures are expressed in minutes, except for the SE expressed in %.

### B. Epoch-by-epoch analysis

The model performance, including single and multi-class metrics, is shown in Table III. The results are calculated on the test set and are expressed in mean ± SD. Per-stage results are reported for Wake, REM, Light, and Deep sleep, and include Sensitivity, Specificity, Accuracy, and F1-score

(mean ± SD) across participants. Global metrics across all sleep stages include overall Accuracy, Balanced Accuracy, Macro-F1 score, and Cohen's $\kappa$, providing a summary of model performance independent of class imbalance. These values reflect the model's ability to classify sleep stages on an epoch-by-epoch basis. In Figure 3, four confusion matrices are shown, comparing our model to the ground truth provided by PSG. Each confusion matrix corresponds to the model results for a different combination of input signals, starting from the model with ACC only. The figure highlights how adding each signal increases each class's performance, including the values for precision, recall, and F1-score. Both absolute and normalized values are shown in the confusion matrices. The main matrix displays the counts and normalized values in brackets for each class. The right column shows recall for each class, the bottom row shows precision, and the bottom-right cell shows the macro-averaged F1-score.

TABLE III: Performance metrics

| Sleep stage | Sensitivity | Specificity | Accuracy | F1-score |
|---|---|---|---|---|
| Wake | 0.59 ± 0.20 | 0.94 ± 0.07 | 0.87 ± 0.07 | 0.60 ± 0.17 |
| REM | 0.55 ± 0.25 | 0.95 ± 0.04 | **0.90 ± 0.04** | 0.53 ± 0.22 |
| Light sleep | **0.76 ± 0.11** | 0.66 ± 0.12 | 0.71 ± 0.07 | **0.72 ± 0.09** |
| Deep sleep | 0.66 ± 0.23 | **0.95 ± 0.05** | 0.89 ± 0.04 | 0.64 ± 0.15 |
| **Global** | **Accuracy** | **Bal. Acc.** | **Macro-F1** | $\kappa$ |
| | **0.69 ± 0.08** | 0.64 ± 0.10 | 0.62 ± 0.11 | 0.50 ± 0.14 |

Performances for single and multi-class sleep stage classification on the test set.

### C. Diagnosis-specific performance analysis

Results per diagnosis are shown in Table IV, including global metrics for the main diagnosis that occurs in our dataset. The normal variants category includes individuals who attended the sleep clinic for evaluation but in whom no sleep disorder was diagnosed. For these participants, some sleep-related findings may appear atypical, but they are considered healthy, and their patterns fall within the range of normal physiological variation.

TABLE IV: Results per diagnosis

| Diagnosis | Accuracy | Bal. Acc. | Macro-F1 | $\kappa$ |
|---|---|---|---|---|
| Breathing disorders | 0.68 | 0.64 | 0.62 | 0.49 |
| Hypersomnolence | 0.72 | 0.67 | 0.62 | 0.52 |
| Normal variants | **0.77** | **0.73** | **0.74** | **0.64** |
| Movement disorders | 0.69 | 0.52 | 0.52 | 0.53 |
| Insomnia | 0.52 | 0.51 | 0.50 | 0.29 |
| Parasomnias | 0.51 | 0.49 | 0.47 | 0.29 |

Mean classification performance for each diagnosis class, computed across all participants in the test set.

## IV. DISCUSSION

The proposed model with ACC, BVP, EDA, and TEMP for four sleep stage classification achieves an overall accuracy of 69%, balanced accuracy of 64%, F1-score of 62%, and $\kappa$ of 50% on the test set. As shown in Figure 1, both EDA and TEMP exhibit correlations with sleep stages; however, preliminary experiments integrating these signals into the model resulted in a decline in classification performance. This observation motivated the development of an alternative preprocessing pipeline specifically tailored to EDA and TEMP,

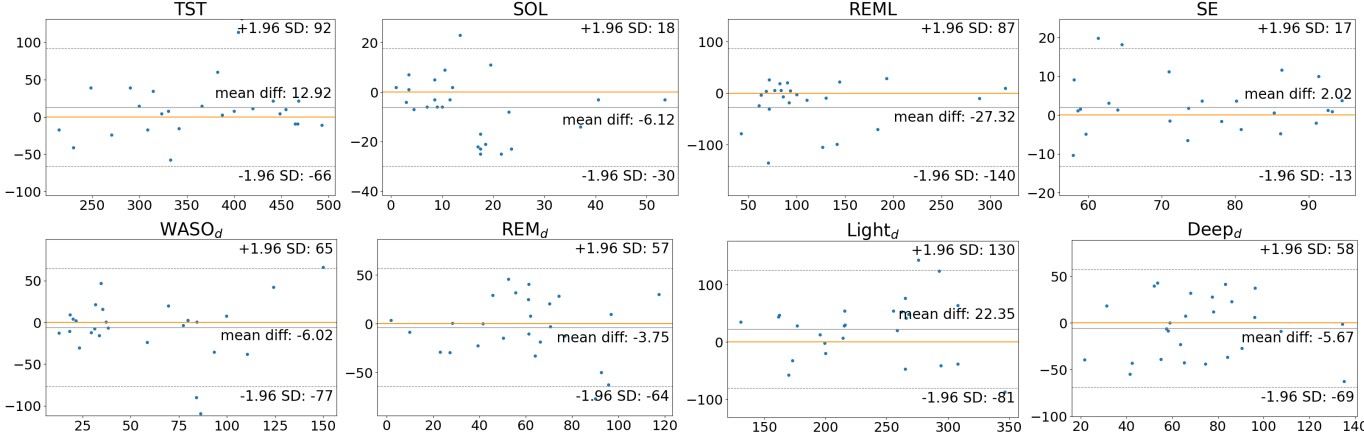

Fig. 2: Bland-Altman plots of sleep measures obtained from our model compared to PSG on the test set. Blue dots represent individual samples, the orange horizontal line indicates zero difference (i.e., perfect agreement), and the gray line shows the mean difference between the two measures.

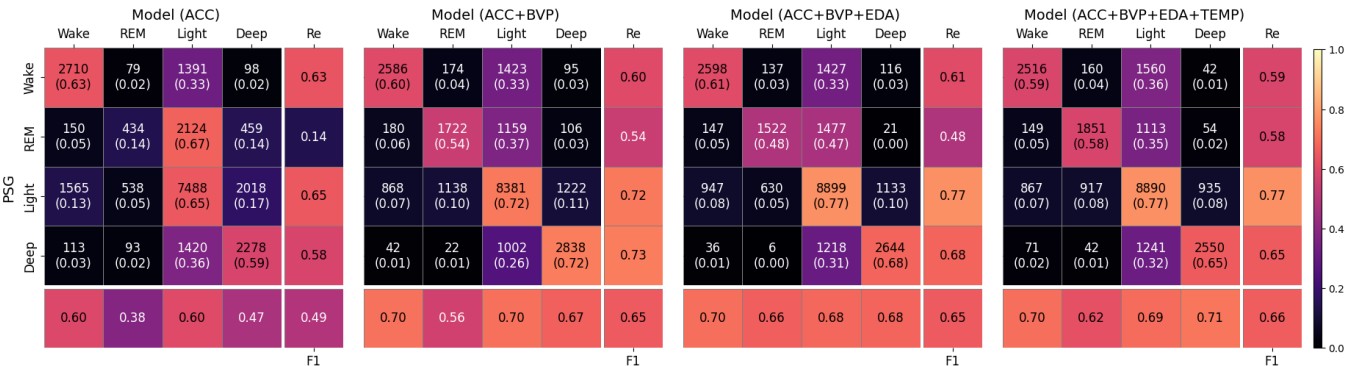

Fig. 3: Confusion matrices illustrate the progressive improvement in classification performance as additional signals are incorporated from a single signal to the complete set of four modalities.

enhancing their contribution to the overall model accuracy. Following the decomposition of EDA into its tonic and phasic components, only the phasic component was retained for further analysis, as it has been shown to exhibit stronger correlations with brain activity during sleep [22]. Specifically, phasic EDA is positively associated with theta waves, predominant during light sleep, and negatively associated with delta waves, characteristic of deep sleep. The decision to use the delta of the temperature signal, rather than the raw signal, stems from the limited effectiveness of spectrogram analysis on slowly varying, low-frequency signals. The raw temperature spectrogram yielded redundant and uninformative patterns due to the signal's low temporal resolution. In contrast, the delta signal introduces mid-range frequency components that are more informative for spectrogram-based deep learning models. Additionally, using the delta attenuates baseline effects and enhances sensitivity to physiologically relevant fluctuations, allowing for improved detection of thermoregulatory patterns, such as peripheral vasoconstriction events typically observed during REM sleep [23]. To provide a comprehensive interpretation of the results, the following sections address the sleep measure and epoch-by-epoch concordance analysis, results per diagnosis, comparison to previous literature, and finally, the study's limitations and directions for future work.

*Sleep measures analysis.* In addition to sleep stage classification, we compared model-derived sleep measures to PSG reference values. As shown in Table II, except for SOL and $Light_d$, no statistically significant differences were observed, suggesting that the model can reliably estimate key sleep architecture metrics in most cases. The discrepancies in SOL may reflect the limited ability of wrist-based sensors to detect quiet wakefulness, a well-known challenge for wearable methods. Similarly, the observed overestimation of light sleep is consistent with classification trends and may stem from the frequent misclassification of wake or REM epochs as light sleep. This effect is likely amplified by the fact that light sleep is the dominant class in sleep patterns, which can bias the model towards overpredicting this stage.

*The epoch-by-epoch analysis* performance of the model across sleep stages highlights both its strengths and limitations. As shown in Table III, the model achieves high specificity for

TABLE V: Summary of related works.

| Work | Model | Signals | Participants | Diagnosis | Accuracy | Bal. Acc. | F1-score | $\kappa$ |
|------|-------|---------|--------------|-----------|----------|-----------|----------|----------|
| Our work | U-Net | ACC + BVP + EDA + TEMP | 127 | Sleep disorders | 69 | 64 | 62 | 50 |
| Olsen et al. (2023) [9] | U-Net | ACC + PPG | 301 | Healthy/BD | 69 | - | - | 58 |
| Li et al. (2021) [18] | CNN + SVM | ACC + PPG | 105 | Healthy/PTSD | 69 | - | - | 44 |
| Silva et al. (2023) [24] | Features RNN | ACC + PPG | 1522 | Healthy/SA | 71 | 72 | - | 56 |
| Song et al. (2023) [25] | SLAMSS | ACT + HRM + HRSD | 808 | Healthy | 72 | - | 73 | - |
| Zhai et al. (2020) [26] | LSTM | ACT + HR | 1743 | Healthy* | 70 | - | 52 | 54 |

Overview of recent four sleep-stage classification approaches, comparing model architecture, input signals, study populations, and evaluation metrics. BD = breathing disorders, SVM = Support Vector Machine, PTSD = post-traumatic stress disorder, RNN = recurrent neural network, SA = sleep apnea, LSTM = long short-term memory, SLAMSS = sequence-to-sequence LSTM for automated mobile sleep staging, ACT = activity, HRM = heart rate mean, HRSD = heart rate standard deviation. *16% with sleep disorders.

most classes, particularly Wake, REM, and Deep sleep, suggesting a strong ability to correctly identify negative instances. However, sensitivity is more variable, with the lowest values observed for REM (0.55 ± 0.25) and Wake (0.59 ± 0.20), indicating that these stages are more prone to underdetection. This imbalance is also reflected in the F1-scores, with REM sleep showing the weakest performance (0.53 ± 0.22), likely due to its shorter duration and physiological overlap with other stages. Light sleep achieved the highest sensitivity (0.76 ± 0.11) and a relatively balanced F1-score (0.72 ± 0.09), although at the expense of lower specificity (0.66 ± 0.12), suggesting a tendency of the model to overpredict this stage. Overall classification metrics (Global Accuracy = 0.69 ± 0.08; Balanced Accuracy = 0.64 ± 0.10; Macro F1-score = 0.62 ± 0.11) indicate moderate agreement with PSG, with a $\kappa$ of 0.50 ± 0.14, placing the model in the range of moderate agreement. These values are promising considering the clinical complexity of the population and are comparable to or better than many prior wearable-based models tested exclusively in healthy cohorts. Finally, Figure 3 shows the impact of progressively adding each signal on the model's performance. Starting with the lowest performance using only ACC, especially in terms of REM and deep sleep detection, the figure shows that the most significant improvement comes from adding BVP. However, the best performance is reached with all signals combined, further improving the detection of REM sleep, the rarest and most challenging stage to classify. This comparison underscores the potential of a multi-modal approach for more accurate and robust sleep stage classification.

*Results per diagnosis.* Most validation studies of wrist-worn wearables have been conducted on healthy populations, typically characterized by consolidated, non-disrupted, and long-lasting sleep periods. As a result, these studies may not fully capture the performance of wearable devices in more complex, real-world clinical scenarios where sleep is often fragmented and influenced by underlying health conditions. In contrast, our study includes participants from a clinical population and extends the validation framework by incorporating analyses stratified by sleep disorder diagnosis, allowing for a more nuanced assessment of device performance across different pathological conditions. The results per diagnosis in Table IV show that the best performance is obtained for normal variants (participants with non-pathological sleep patterns), for which an overall accuracy of 77%, balanced accuracy of 73%,

F1-score of 74%, and $\kappa$ of 64% are achieved. The lowest performance is observed in participants with insomnia and parasomnias, likely due to the limited number of subjects with these diagnoses and the greater difficulty in accurately detecting their sleep patterns. Larger, balanced datasets are needed to develop diagnosis-specific modeling. However, our model shows that adding signals correlated to sleep provides performance comparable to SOTA [24] even with challenging datasets.

*Comparison to Literature.* Table V compares our model to previous work using wearable data for sleep staging. Olsen et al. [9] applied U-Net to a largely BD cohort; Li et al. [18] used CNN-SVM on PTSD participants; Silva et al. [24] used RNNs with hand-crafted features on a mostly healthy sample; Song et al. [25] applied LSTM (SLAMSS) on ACT and HR-derived metrics; Zhai et al. [26] used LSTM on ACT and HR with limited sleep disorder representation. In contrast, our study focuses on a diagnostically diverse clinical population with fragmented sleep and limited data (127 participants), posing greater modeling challenges. Despite this, our multimodal approach achieves performance comparable to existing work, underlining its robustness and clinical relevance.

*Limitations and future work.* Conducted in a clinical setting, our study may be affected by less standardized manual scoring due to time constraints and variable scorer expertise. The modest sample size and unbalanced diagnostic representation may limit generalizability and affect performance for under-represented groups. To improve clinical applicability, the U-Net model could be extended to handle full-length, variable-duration inputs without truncation. Diagnostic information could be integrated into training via weighted loss or multitask learning. Addressing class imbalance through oversampling and stratified training may further improve generalizability across diverse patient populations.

## V. CONCLUSIONS

We demonstrated the feasibility of deep learning-based sleep staging from wrist-worn sensors in a clinically complex population. Unlike prior studies on healthy individuals, our model addresses real-world variability in sleep disorders and performs competitively. This work supports wearable-based, personalized sleep monitoring as a scalable alternative to PSG and highlights the need for adaptive AI models that integrate diagnostic context, paving the way toward patient-centered care in sleep medicine.

## ACKNOWLEDGMENT

The authors thank the study nurses and sleep technicians at the Sleep-Wake-Epilepsy Center for making this project possible and the data manager, Dr. Julia van der Meer, for her help in collecting the PSG data. The authors thank the members of the Gerontechnology Rehabilitation Group, ARTORG, University of Bern, for contributing to the implementation of this study and the acquisition of the wearable sensor data.

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
