# OpenReview forum: "Sleep Stage Classification from Wristband Sensor Data in Patients with Sleep Disorders"
_IEEE.org/EMBS/BHI/2025/Conference — BHI 2025_

### Official Review · Reviewer_55wh · 2025-06-28
**Sleep Stage Classification from Wristband Sensor Data in Patients with Sleep Disorders**

**Confidence:** 5
**Clarity Of Writing:** excellent
**Clinical Significance:** excellent
**Methodological Novelty:** fair
**Overall Rating:** 7

**Experiments And Results:**

great

**Questions For The Authors:**

The use of multimodal models comes with the assumption that individual signals are not sufficient to obtain good performance. Why is the performance for the model not compared with individual modality on the same dataset to show improved performance?

**Strengths:**

The paper is clearly written, providing relevant information at appropriate points, and the results are effectively communicated.

**Summary Of The Paper:**

This study presents a multimodal, AI-driven approach to sleep stage classification using data from the Empatica E4 wristband in individuals with sleep disorders. By integrating acceleration (ACC), blood volume pulse (BVP), electrodermal activity (EDA), and skin temperature (TEMP), the U-Sleep neural network was trained and evaluated. Results show strong agreement with PSG for most sleep metrics. The approach demonstrates the potential of wearable multimodal data for personalized and remote sleep monitoring, contributing to precision sleep health.

**Weaknesses:**

The study does not provide a comparative analysis to evaluate the individual contribution of each signal modality to the model's performance, which limits understanding of their relative importance.

The model architecture is neither described in detail nor illustrated with a diagram, which hinders the reader’s ability to fully understand the approach and limits the reproducibility of the study.

---

### Official Review · Reviewer_pyN3 · 2025-07-17
**Innovqtive adaptable multimodal sleep stage classification tool**

**Confidence:** 5
**Clarity Of Writing:** good
**Clinical Significance:** good
**Methodological Novelty:** good
**Overall Rating:** 7

**Experiments And Results:**

great

**Questions For The Authors:**

* in Tables II and III, please highlight best performance using different color/font format
* please describe in short programming platform and computing systems characteristics for analysis multi-source signals
* please provide quantitative assessments on time-efficiency of proposed monitoring framework. Can it be considered a real-time system? What is the estimated TRL of the overall prototype?
* in Fig. 3 please ensure that all rows in Confusion Matrices sum up to 100%
* a few syntax errors need to be corrected
* indicative illustration of monitoring prototype would be useful

**Strengths:**

* cost-effective and flexible solution
* analysis of multiple types of information
* validation on real-world data and under different metrics
* up-to-date analysis of SOTA, based on numerous references
* extended discussion Section with comparative analysis and description of limitations and future improvement

**Summary Of The Paper:**

This study presents an innovative tool capable of incorporating analysis of diverse physiological signals into a personalized model for individuals with sleep disorders. An interesting contribution lies in the feasibility, portability and adaptability of proposed framework, since it is  based on cost-effective wearable devices.

**Weaknesses:**

* as already pointed out by the authors "Larger, balanced datasets are needed to develop diagnosis specific modeling". However, preliminary results look quite promising
* understanding of monitoring system prototype by the reader is not totally feasible without illustration/figure/flowchart/image of the operation in practice

---

### Official Review · Reviewer_mqtc · 2025-07-18
**Good - however it has some major weaknesses**

**Confidence:** 3
**Clarity Of Writing:** fair
**Clinical Significance:** good
**Methodological Novelty:** fair
**Overall Rating:** 2

**Experiments And Results:**

poor

**Questions For The Authors:**

What are the clinical implications of these results? If a doctor wants to use this tool for sleep disorder diagnosis, how well can the model's performance be explained to doctors in non-technical terms?
Were there any outliers? How does the model treat them? What is its impact on clinical outcomes?
In order to improve the ecological validity of the dataset, the author can provide a discussion around the data quality of the wearable device data, for example, outliers, signal-to-noise ratio, and any corrupted signals that were filtered.
Are there any clinical raters involved in the study to evaluate the participants' sleep quality?
Were there any surveys done to validate the qualitative sleep data along with the model's performance?

**Strengths:**

The dataset consists of a good number of participants, 127, for training the CNN model.
Multimodal signals of the dataset with a wrist-based device are a strong suit for this dataset.
Participant demographic information is well presented with clear train-test splits.
A thorough table presenting a comparison with state-of-the-art is provided

**Summary Of The Paper:**

Overall, the paper presents an evaluation of the U-Sleep model on data collected from 127 participants for clinically relevant individuals. Unlike previous work, which was primarily focused on healthy populations, their approach centers on evaluating wearable-based sleep staging in clinically relevant individuals characterized by heterogeneous, often disrupted sleep patterns.

**Weaknesses:**

Introduction
The paper is a bit unclear on what the main contribution is. Two sections differ: In the Methods section, the authors refer to “The main contributions of our work include adapting the network presented in [10], experimenting with various signal combinations, identifying the optimal parameters, and conducting a comprehensive model evaluation similar to that performed on the other devices.” However, in the introduction, the authors mention, “Unlike previous work, primarily focused on healthy populations, our approach centers on evaluating wearable-based sleep staging in clinically relevant individuals characterized by heterogeneous, often disrupted sleep patterns. This distinction aligns with the core principles of precision health, where models and tools must adapt to the specific needs of diverse patient groups. By leveraging multimodal sensor data, including ACC, BVP, EDA, and skin TEMP, we investigate the potential of Artificial Intelligence (AI) to learn individual-specific physiological patterns that reflect distinct sleep architectures.”
If adapting U-sleep is the main contribution, the paper must introduce a figure on the updated neural network architecture. If the clinical population is the main context of the paper, the paper must include a discussion on why the dataset is imbalanced and how the authors applied different strategies to overcome this.

Materials and Methods
Not enough details on why they chose a specific way to process the data. While the citations are there, sometimes it's hard to relate to the cited paper.
Input and output layers of the network need to be defined based on the dataset size.
“The wearable data were trimmed between the PSG lights-off and lights-on times.” This needs more details on why.- citation not given or add explanation depending on
Why was a 300-second window chosen?
What is the clinical relevance of having all these signals to classify sleep ACC, PPG, EDA, and TEMP. Authors can mention the relevance of these biomarkers.

Evaluation
For Fig. 1, the line plot resolution and color shading can be improved to improve readability. This is an important figure of the paper as it showcases the dataset signals. Labels and legends are not clearly defined.
In Fig. 3, the confusion matrix numbers are not visible clearly.
While Table 4 discusses the results per diagnosis, the dataset is heavily imbalanced. How did the authors take this into consideration? The accuracy shown can be skewed due to the imbalance in the dataset. More discussion is needed.
equations in a paragraph are hard to read. Equation can standout by themselves.
TABLE IV. A deeper discussion of these results is needed, as the main contribution of the paper is as follows, as per the introduction: “Unlike previous work, primarily focused on healthy populations, our approach centers on evaluating wearable-based sleep staging in clinically relevant individuals characterized by heterogeneous, often disrupted sleep patterns.” Why is the F-score for individuals with Parasomnias the lowest?
Wondering if there are any results on TABLE IV: Results per diagnosis about the performance on different classes, such as Wake, REM, and Light sleep.
Comparison to Literature: Authors should provide a strong argument as to why their accuracy is lower than Silva et al. (2023) [23], Song et al. (2023) [24], Zhai et al. (2020) [25]
Any results on models' evaluation on PSG SOMNOscreen® HD Somnomedics data? Especially since the authors mentioned in conclusion that this is an alternative to PSG, however, I didn't see any results comparing the two.

Minor comments:
Overall, the reference format should follow IEEE style.
Language in the methods section can be improved to produce clarity on why certain methods were chosen.
Figure quality and resolution can be improved.
Authors could have utilized the entire eight pages, while some of sections need more clarity and details.

---

### Official Review · Reviewer_e4jp · 2025-07-18
**Sleep Stage Classification from Wristband Sensor Data in Patients with Sleep Disorders**

**Confidence:** 4
**Clarity Of Writing:** good
**Clinical Significance:** good
**Methodological Novelty:** good
**Overall Rating:** 6

**Experiments And Results:**

good

**Questions For The Authors:**

Did the authors ensure that the dataset was properly split, with distinct segments from each patient assigned to the training, validation, and test sets, thereby preventing any data leakage between sets?

There are formatting errors present, where several figures have duplicate captions—one above the figure and another below. These should be corrected, with the caption placed consistently to adhere to standard formatting guidelines.

The lack of an independent dataset for validation limits the ability to assess the model's performance on unseen data.

**Strengths:**

The multimodal strategy enables a more holistic and robust analysis of sleep patterns by capturing diverse physiological responses.

The multiclass classification framework allows the model to distinguish among various sleep stages and conditions, enhancing the granularity and clinical relevance of the results.

**Summary Of The Paper:**

This study presents an AI-based approach for stratifying sleep disorders using a U-Net architecture trained on multimodal physiological signals. The dataset includes Acceleration (ACC), Blood Volume Pulse (BVP), Electrodermal Activity (EDA), and skin Temperature (TEMP), all collected using the Empatica E4 wristband. By leveraging these diverse data streams, the model aims to provide a comprehensive and accurate classification of sleep stages.

**Weaknesses:**

While other AI-based models can be used for comparison, the results clearly demonstrate the superior performance of the U-Net model in this study. A direct comparison with previous work highlights the advancements that U-Net can achieve in sleep disorder stratification.